# Does Connected Health Technology Improve Health-Related Outcomes in Rural Cardiac Populations? Systematic Review Narrative Synthesis

**DOI:** 10.3390/ijerph19042302

**Published:** 2022-02-17

**Authors:** Matthew James Fraser, Trish Gorely, Chris O’Malley, David J. Muggeridge, Oonagh M. Giggins, Daniel R. Crabtree

**Affiliations:** 1Division of Biomedical Sciences, Centre for Health Science, University of the Highlands and Islands, Inverness IV2 3JH, UK; daniel.crabtree@uhi.ac.uk; 2Department of Nursing and Midwifery, Centre for Health Science, University of the Highlands and Islands, Inverness IV2 3JH, UK; trish.gorely@uhi.ac.uk; 3Library, Centre for Health Science, University of the Highlands and Islands, Inverness IV2 3JH, UK; chris.omalley@uhi.ac.uk; 4School of Applied Sciences, Edinburgh Napier University, Edinburgh EH11 4DE, UK; d.muggeridge@napier.ac.uk; 5NetwellCASALA, Dundalk Institute of Technology, A91 K584 Dundalk, Ireland; oonagh.giggins@dkit.ie

**Keywords:** cardiovascular disease, digital technology, wearable technology, rural, connected health

## Abstract

Individuals living in rural areas are more likely to experience cardiovascular diseases (CVD) and have increased barriers to regular physical activity in comparison to those in urban areas. This systematic review aimed to understand the types and effects of home-based connected health technologies, used by individuals living in rural areas with CVD. The inclusion criteria included technology deployed at the participant’s home and could be an mHealth (smart device, fitness tracker or app) or telehealth intervention. Nine electronic databases were searched across the date range January 1990–June 2021. A total of 207 full texts were screened, of which five studies were included, consisting of 603 participants. Of the five studies, four used a telehealth intervention and one used a form of wearable technology. All interventions which used a form of telehealth found a reduction in overall healthcare utilisation, and one study found improvements in CVD risk factors. Acceptability of the technologies was mixed, in some studies barriers and challenges were cited. Based on the findings, there is great potential for implementing connected health technologies, but due to the low number of studies which met the inclusion criteria, further research is required within rural areas for those living with cardiovascular disease.

## 1. Introduction

Cardiovascular disease (CVD) is the leading cause of death globally, accounting for 31% of annual deaths [1]. Older adults living in rural areas, who have lower socioeconomic and education levels, are more likely to live alone and have greater levels of disability and multimorbidity than those in urban areas [2]. These factors contribute to an increased risk of developing CVD [3]. Individuals living with CVD need to conduct regular physical activity to effectively manage the condition [4]. Self-management has also become an area of interest within CVD research, with recent studies examining methods to improve this [5]. However, rural residents also have increased barriers to attending regular healthcare and physical activity with fewer places to be active indoors and outdoors [6]. Gilbert et al. (2019) [7] found environment, accessibility, reduced resources, infrastructure, weather and reduced social support to be barriers to physical activity for rural residents. Barriers to attending regular healthcare include a paucity of services, a lack of trained specialists, transport and finance [8]. The American Heart Association (AHA) published a ‘call to action’ to reduce geographic health disparities, emphasising the need for more research within rural cohorts [9]. 

Individuals living in rural areas are more likely to experience cardiac issues and CVD in comparison to those from urban areas [10]. Reasons include dietary choices, reduced access to healthcare services, increased smoking status, and lower physical activity levels than those in urban areas [10,11]. For individuals already living with a form of CVD, cardiac rehabilitation (CR) is regarded as essential [12]. In the UK, 100,000 patients attend CR, approximately 50% of those who are eligible [13]. The attendance of rural patients is even lower [11] with one report showing that rural participants were 71% less likely to utilise CR [14]. Barriers to attending CR for patients living in rural areas have been shown to be primarily related to geography. Barriers highlighted in the literature include distance to the class, transport access, parking costs [15], infrastructure in terms of the quality of roads and poor weather conditions [16].

Technological advances have allowed the development of CR conducted at the patient’s home, to increase accessibility [17]. Previous reviews have been conducted into the effectiveness of technology within rehabilitation settings [18,19], though there has been little consideration of rural patients. Understanding what effect technologies can have for rural residents living with CVD is important due to the range of barriers associated with physical activity and health care accessibility. Various terms have been used to describe the types of technology utilised in health settings. The term mHealth has been defined by the WHO as the delivery of medical services via devices such as smartphones, tablets, or wearable monitoring devices, and this has been expanded to mobile apps, social media, and GPS [20]. In home settings, smartphone-based interventions have been shown to improve attendance and completion of CR programmes [21]. Another form of technology frequently employed within health care settings are ‘digital health interventions’, which is an overarching term used by Wongvibulsin et al. (2021) [18] to describe ‘telemedicine, telehealth and eHealth.’ Such methods allow patients to communicate with healthcare professionals through mediums such as email, audio, or video [22]. Telehealth also allows the delivery of online exercise programmes at distance and Rawstorn et al. (2016) [23] found CR telehealth interventions to be ‘at-least’ as effective in improving CVD risk factors and functional capacity in comparison to centre-based CR.

An integrative review by Field et al. (2018) [24] on CR services within rural patients was conducted and found several factors related to access and referral. Such factors were related to patient education during hospital stay improvements to the referral and discharge process and a need for improved accessibility (including home-based exercise alternatives). However, the review excluded studies around centre and home-based technology. Moreover, researchers have outlined that further research into rural e-health is required as it develops [25]. This is supported by Jaana and Sherrard (2019) [26] who note that identifying the effects of home-based technology on rural residents allows understanding of which forms of technology are most suitable and which can increase uptake and the effectiveness of CR programmes. They also highlight the paucity of research exploring telehome monitoring technologies within rural populations. It is therefore important to assess what forms of home-based technology have already been deployed and what the effects have been, to understand which have the greatest effects.

Aims and objectives

The aims of this review are threefold:To identify the types of home-based technologies that have been used with rural residents living with CVD or a cardiac condition.To examine the acceptability, adherence, and usability of the home-based technologies for cardiac patients living in rural areas.To examine whether home-based connected health technologies improve health-related outcomes for cardiac patients living in rural areas.

## 2. Materials and Method

The current systematic review was performed in accordance with the Preferred Reporting Items for Systematic Reviews and Meta-Analyses (PRISMA) guidelines [27]. This systematic review is registered with PROSPERO (CRD42021248518). All database searches were conducted in June 2021. The search was comprised of three areas: cardiovascular disease, home-based technology, and rural participants. Initially, the review was focused on the effects of technology delivered cardiac rehabilitation for rural patients, however, there was no research which met the criteria, and therefore this was expanded to rural individuals living with a form of CVD or heart condition using home-based technology. The eligibility criteria are described below. The current systematic review is a narrative synthesis including quantitative, qualitative, and mixed method study designs. This method was selected due to the heterogeneity of study designs, outcome measures and technologies. The narrative synthesis aimed to develop a preliminary synthesis for the effects of the included studies and understand why specific findings were seen [28]. This analysis examined similarities and differences within the included studies of the current systematic review. 

Eligibility Criteria

Inclusion Criteria

The inclusion and exclusion criteria were developed using the PICOs framework designed for systematic reviews [29].

PICOs Criteria

Population: Patients (>18 years old) living with a cardiac issue or CVD in rural areas (due to the wide range of definitions of rural areas in the literature, the following classification was used, “*each community was at the nonmicropolitan level (less than 9999) or micropolitan level (10,000 to 49,999)*” [7] (p. 2).Intervention: Implemented an mHealth or wearable technology intervention (smartphone, smartwatch, fitness tracker, pedometer, accelerometer) or telehealth intervention (programme delivery, support, counselling, education) in a home-based setting. The definition of ‘home-based’ was the same as used in the study by Blair et al. (2011) [30] (p. 3), “*rehabilitation delivered at the patients’ home or in a local, non-hospital location*.” With regard to the types of interventions considered within the current systematic review these included exercise, feasibility, cardiac rehabilitation, education/counselling, and technology experience.Comparison: All comparison types were considered within the current review to capture the full range of interventions (RCT, comparisons to groups without technology, comparison to other types of technology, and comparison to control groups)Outcomes: Various outcome measures were considered (uptake and adherence to the technology, completion, usability and acceptability of technology, physical activity levels, psychological and physiological outcomes, healthcare utilisation and hospitalisations, and economics).Study design: Both randomised as well as non-randomised studies were eligible for inclusion. There were no restrictions on sample size or follow-up duration.

### 2.1. Search Strategy 

Nine online electronic databases were systematically searched; CINAHL, Cochrane Library, Embase, Medline/Ovid, Scopus, SPORTDiscus, Web of Science, PubMed, Google Scholar. Grey literature was also searched by two authors (M.J.F, C.O). The following websites were searched for grey literature; National Institute for Health Research; Health Data Research UK; General Medical Council; Care Quality Commission; UK Department of Health and Social Care; NHS Research Scotland; Ethos; ISRCTN Registry; DART-Europe; Open Grey; Royal College of Emergency Medicine; Royal College of Surgeons Edinburgh; University of the Highlands and Islands. The British Library were contacted when there was limited access to specific papers. Databases were searched for literature published from January 1990–June 2021. This search date was selected because it was unlikely that connected technologies would be reported on prior to this date, and this timeframe was used by Wongvibulsin et al. (2021) [18], who investigated digital health interventions for CR. Articles had to be in or translatable into English. Various search strategies were implemented for each of the databases with help from an information specialist (C.O), and all search strategies are provided in the Appendix A. Appropriate synonyms were used across each database and the various sources used to search the grey literature materials, to provide a wide ranging and comprehensive search of the literature. MeSH terms were used for several phrases central to the outcomes of the study. All database searches are included in Appendix A. Additionally, a citation search was conducted on the reference list of one previous systematic review to ensure any relevant articles were not missed. 

### 2.2. Study Selection Process

Two authors (M.J.F, T.G) independently screened titles and abstracts. The authors had 96.9% agreement on included studies. Full text screening was completed by two researchers (M.J.F, D.R.C). Disagreement was found for only one study. Disagreements were resolved by consensus or by a third researcher (D.R.C, T.G).

### 2.3. Data Extraction and Study Quality

One author (M.J.F) extracted data from each of the studies which met the eligibility criteria. This was checked and confirmed by a second researcher (T.G). The following data/headings were extracted; title, year of publication, the author(s), location, keywords, aims, number of participants, participant characteristics (age, gender, cardiac condition), study design, duration, type of technology and intervention, adverse events, outcome measures, results, future recommendations. If any disagreements were found, they were resolved by consensus or a third researcher (D.R.C). 

### 2.4. Study Quality and Risk of Bias 

Two researchers (M.J.F, D.R.C) independently examined the quality and risk of bias for each of the included studies using the Downs and Black quality scale [31]. The quality assessment tool was developed to evaluate the methodological quality of healthcare interventions, increasing the validity for the present systematic review [31]. Moreover, the tool can assess both randomised and non-randomised study designs, which provides a rationale for its selection. The Downs and Black quality index is comprised of 27 ‘yes–no’ questions, categorised into 5 sections; quality, external and internal validity, bias, confounding and selection bias, and power [31]. It has been previously validated showing good levels of both test–retest (r = 0.88) and inter-rater (r = 0.75) reliability. Five studies were scored by two authors (M.J.F and D.R.C) independently, and any conflicts of opinion were discussed and agreed upon. The modified version which we employed in this study therefore has a maximum score of 28. Scoring was completed via the current categories “excellent” (24–28 points), “good” (19–23 points), “fair” (14–18 points) or “poor” (<14 points) [32]. Any discrepancies in assessment were resolved by discussion between the two researchers. A third researcher (T.G) was used if no consensus could be reached to resolve the discussion. 

## 3. Results

### 3.1. Search Outcome 

The initial database search resulted in a total of 1657 articles. This was reduced to 1601 after removing duplicates. The titles and abstracts of studies were screened, and 1394 were then removed due to not meeting the eligibility criteria. The full text of 207 articles were screened. After evaluation, 202 out of the 207 articles were excluded due to the following reasons: Wrong population, wrong type of technology, wrong outcome measure used, wrong study design, wrong type of publication and language. This resulted in a total of five studies which met the criteria for the systematic review. Studies were screened from citation and grey literature searches, however, studies that appeared to meet the criteria were excluded due to technology used away from the home and including conditions outside of CVD and cardiac issues. Figure 1 presents the search results within the current study.

### 3.2. Description of Studies 

The aim of the current systematic review was to understand what forms of technology have been deployed and the effects of home-based technology interventions within rural cardiac patients. Information which was extracted from the studies is provided in Table 1 and Table 2.

### 3.3. Study Quality

Of the 5 studies, 2 were fair quality, 2 were of good quality, and 1 was excellent. The quality of the included studies is presented below in Table 3.

### 3.4. Outcomes

#### 3.4.1. Participants

A total of 603 participants were included in the review. All participants resided in a rural area as per the study inclusion criteria. Four studies investigated the use of home-based technology on patients with heart failure [33,34,35,37]. ther heart conditions included within the current systematic review were myocardial infarction, coronary artery bypass graft and angioplasty [36]. Studies were conducted in the United States [33,34,35,37], and Canada [36]. Typically, participants were older adults (>65 years old), but ages ranged from 18 –> 85 years old. 

#### 3.4.2. Interventions

The duration of the interventions ranged from 12 weeks to 16 months. Study designs were both randomised and non-randomised. Implemented study designs included both quantitative and qualitative approaches. One study used a mixed method design [33]. Secondary analysis and programme evaluation was used by two studies [34,35]. One study used a randomised controlled trial [36]. Huntington et al. (2013) [37] used a non-randomised design to investigate telehealth and education.

#### 3.4.3. Types of Technology

There was a mixture of both wearable technology and telehealth utilised. Only Young et al. (2017) [34] used wearable technology, in the form of a waist worn tri-axial accelerometer (Actigraph). Lefler et al. (2018) [33] used digital home equipment (Cloud DX connected health kit), which was comprised of an Android health tablet, Bluetooth paired weight scale and a blood pressure wrist monitor. The ‘in-home equipment’ group were issued with a digital blood pressure wrist monitor and digital scales which did not transmit data. Lear et al. (2014) [36] investigated virtual cardiac rehabilitation, which included online forms (risk factors and lifestyle), one-to-one chat sessions, weekly education, and expert led group chat sessions. Participants were also issued with heart rate monitors (Polar s610i) and blood pressure monitors (Lifesource, UA779). Riley et al. (2015) [35] evaluated the Care Beyond Walls and Wires programme. This was a remote monitoring service which included a wireless peripheral device to measure bodyweight, blood pressure, heart rate and pulse oximetry, transmitted to a Motorola Droid X2 smartphone and then to a nursing team. In the study by Huntington et al. (2013) [37], the researchers used a telehealth intervention to deliver educational sessions to heart failure patients. In total, there were eight different forms of technology across the five studies (blood pressure monitor, accelerometer, Bluetooth bodyweight scale, heart rate monitor, tablet, pulse oximeter, internet, telephone/smartphone). 

#### 3.4.4. Acceptability, Adherence, and Usability

There was a range of outcome measures used within the included studies. As seen within similar technology reviews, participants’ levels of acceptability, usability and adherence were assessed within four out of the five included studies. Overall, participants reported reasonable levels of acceptance towards the various types of connected health technologies. Young et al. (2017) [34] required participants to wear an accelerometer for seven consecutive days for at least 8 h per day for 6 months. They found low compliance rates, noting only 54% achieved 100% compliance. Participants wore the accelerometer during weekdays for an average of 15.7 ± 3.3 h and 15.8 ± 3.7 h for weekends, but reported several barriers and challenges to wearing the Actigraph. Significant factors associated with wear time were age, insulin use, being hospitalized, and B-type natriuretic peptide value. The researchers found acceptable levels of reliability of the Actigraph (ICC => 0.8) for calories, active minutes, and moderate intensity activity during habitual levels of physical activity. Lefler et al. (2018) [33] found 36% of participants experienced anxiety towards the mobile health technology, citing nervousness or fear. Contrastingly, participants displayed 80–92% satisfaction regarding communication, 100% adherence in relation to ‘not forgetting to monitor themselves’ and 12/15 participants rated technology usability at above 90/100 (84.2 mHealth group, 95.8 home equipment group). In terms of feasibility 93% of participants agreed that the equipment was easy to use. Riley et al. (2015) [35] found participants rated the programme 4/5 for all satisfaction and usability items except for equipment being easy to use and overall experience for 1 patient. 

#### 3.4.5. Health-Related Outcomes 

Health service utilisation was examined in four out of five of the included studies. These ranged from emergency department visits [33,35,36] to 30-day all cause readmissions [37]. Riley et al. (2015) [35] found reductions in healthcare utilisation, but no significant differences between the home-based technology group versus a ‘matched’ group who did not take part in the remote monitoring intervention but were hospitalised during the project and a ‘declined’ group who were asked to take part in the study but declined. They did, however, find that the average costs of hospital charges decreased by 72% from the 6 months prior to enrolment in the technology group. Similar economic savings were found by Huntington et al. (2013) [37]. The authors found a 128% return on investment within the technology enrolled group. Physical and physiological outcome measures were also assessed in the study by Lear et al. (2014) [36]. The study found participants randomised to the virtual cardiac rehabilitation condition significantly improved their performance in an exercise capacity treadmill test in comparison to the usual care group by 45.7 (95% CI = 1.04–90.48, *p* = 0.045) seconds. Cholesterol (*p* = 0.026) and low-density lipoprotein (*p* = 0.022) were lower in the virtual cardiac rehabilitation group. 

#### 3.4.6. Qualitative Approaches

In three of the studies, a qualitative approach was used in the form of interviewing participants. In the study by Lear et al. (2014) [36], virtual cardiac rehabilitation participants undertook a semi-structured interview at the end of the intervention to assess satisfaction and attitudes. The data was analysed using a qualitative descriptive approach. Participants found it to be an accessible and convenient way to receive healthcare. They also noted improvements in self-managing their condition, motivation, disease awareness and confidence. Participants reported that overall, they felt as if their health had improved as a result of the programme. Lefler et al. (2018) [33] conducted semi-structured interviews with participants to gain an understanding of feasibility and satisfaction, using a content analysis approach to analyse the data. The results highlighted four key themes related to participants experience of the intervention. The four themes were: traditional communication and engagement with health care providers; home monitoring with technology; awareness of the importance of self-monitoring and management; and persistent health problems. Qualitative responses demonstrated both good and bad aspects of provider communication such as the time taken to respond and with only one patient utilising the communication portal. Participants in the two technology groups noted that they found the technology helpful for managing their condition, and this created confidence, security, and alliance with the research team. The two intervention groups also highlighted that their understanding of how and why to monitor their condition was increased and they would continue to do this into the future as a result of the routine and reminders to continue monitoring their symptoms. Young et al. (2017) [34] conducted individual interviews and focus groups on participants’ experience of wearing the accelerometer, using an inductive analysis process. They found that barriers including trouble wearing it during holidays or unexpected events such as travelling, visiting hospital or when feeling ill, skin problems when wearing the device, fit problems, difficulty taking it on and off and interference with self-care or scheduled medication. Lefler et al. (2018) [33] also used questionnaires to collect information around symptoms, medical events, and behaviour adherence. The study found that from baseline participants in either the mHealth group or home equipment improved their monitoring by 50%. MHealth group participants also felt less discouraged with their health from baseline and felt more empowered to self-care following the intervention.

## 4. Discussion

The current systematic review investigated the types and the effects of home-based, connected health technology, deployed with CVD and cardiac patients living in rural areas. Harrington et al. (2020) [9] found that CVD mortality is worse in rural areas in comparison to urban. The current review found both wearable technology and telehealth had mixed effects on a range of health-related outcome measures. To the authors’ knowledge, only one previous systematic review has considered the effects of technology on CVD within rural cohorts. An integrative research review by Graves et al. (2013) [25] investigated the role of telehealth in rural populations living with heart failure. The review found that telehealth increased access to services and improved health management within heart failure patients. However, from the 12 included studies in the review, only four were from rural populations and thus researchers have highlighted a lack of research on technology in CVD patients living in rural areas [25,26]. The current systematic review built on the review by Graves et al. (2013) [25], though, unlike Graves et al. (2013) [25], the current review did not exclusively include heart failure studies and instead considered all forms of CVD and cardiac issues. The outcomes of the included studies were heterogenous, which did not allow a meta-analysis to be conducted. 

### 4.1. Types of Technologies 

The aim of the current systematic review was to firstly understand what types of technologies have been used within rural patients living with a cardiac condition. Connected health technology was categorised into two forms: telehealth and wearables. Only one study utilised a form of wearable technology despite the widespread uptake of such technologies by the public [38]. A previous systematic review by Akinosun et al. (2021) [39] considered the effects of digital health interventions (phone apps, internet, wearable sensors) within CVD patients. The study found that these technologies may improve healthy behaviours such as medication adherence, but the researchers did not find evidence to support the effects on unhealthy behaviours. Similarly, the current review found only one study which used wearable technology as an intervention [34], and this used secondary analysis methodology to assess the feasibility of a hip-worn accelerometer. Based on the findings of both reviews, more research is required to investigate the role that wearable technology can have on improving both health behaviours and health outcomes related to CVD. With wearable sensors becoming more available, it is likely that as these evolve, they will be more frequently used within cardiac patients [40]. Examples include developments in the Apple Watch, which has seen the ability for these devices to conduct ECG assessments, which could be useful within a cardiac population [41]. However, before such forms of technology will be accepted within healthcare settings, companies need to demonstrate the accuracy and reliability of such devices [42].

Telehealth was the most frequently utilised technology within the included studies, being investigated in four studies. Such forms of technology are now frequently used in healthcare settings to identify and monitor disease, communicate, educate, and promote self-monitoring [43]. Telehealth technologies used at the participants’ home included telephone education and coaching, virtual cardiac rehabilitation, and technologies to measure physical health parameters. Due to the barriers to physical activity and accessibility to health care services experienced by rural patients, utilising technologies for communication and health assessment at a distance are perhaps most warranted, and this is reflected in the findings of this systematic review. Other forms of home-based telehealth which were not seen within the current review which have been previously used in CVD studies included videoconferencing software that can deliver physical activity, online group sessions and smartphone applications, which provide several uses. In only one study was virtual cardiac rehabilitation considered for rural patients, and this was delivered via an internet webpage that displayed tasks and information around their condition [36]. As can be seen from the included studies, telehealth appears to be one method to improving health outcomes and service utilisation for those living with CVD in rural areas. However, further research is needed to confirm this and what types of telehealth are most effective. 

### 4.2. Acceptability, Adherence, and Usability

A second aim of the systematic review was to examine the acceptability, adherence, and usability of the connected health technologies. Understanding if and how participants interact with the technology is in many cases more important than the effects of the technology. If participants do not adhere to the technology, then little effect will be seen. Young et al. (2017) [34] reported adherence to an accelerometer of 45–56% across 6 months, which was deemed low. Qualitative responses revealed discomfort, skin issues (rashes, increased perspiration, and trapped moisture), fit and interference with daily activities as reasons for non-compliance with the technology. Such reasons have been reported when using wearable sports technology in a healthy sample [44]. Thus, it appears for technology designers that finding innovative methods to increase the comfort, size, and usability of such technology may be key in encouraging people to adopt it. In the one study [36] which investigated telehealth in the form of virtual cardiac rehabilitation, interview responses highlighted that the programme was an accessible method to access healthcare and specifically communicate with health professionals. Moreover, participants cited increased levels of motivation, awareness and confidence, which all contributed to improved health and self-management behaviours. This is perhaps achieved through telehealth via ease and quick access to health professionals, increasing self-efficacy and knowledge around their condition. Telehealth technologies facilitate quick and effective communication with healthcare staff and therefore is especially important within rural populations where travel to healthcare facilities is challenging. 

Most participants included within the five studies were above 65 years old, which may explain why in some studies low compliance rates or issues with the different forms of technology were found. Older adults have been shown to experience issues using and accepting technology due to technology literacy [45], costs [46], self-efficacy [47], attitudes [48] and anxiety [49]. Investigating acceptability, adherence, usability, or feedback in relation to the technology was deemed important, as the sustained utilisation of connected health technologies can increase longer-term engagement, motivation, and physical activity levels [50]. Lefler et al. (2018) [33] found older adults experienced anxiety towards using technology, citing nervousness or fear prior to the intervention, which did not improve. Despite this, participants in the technology groups displayed high levels of adherence, feasibility, and usability towards the technology. This demonstrates that participants’ anxious preconceptions are perhaps exaggerated in comparison to what their actual abilities are. Understanding participants’ experience of technology is also a key aspect of whether it will be adhered to long-term, which is essential to successfully altering behaviour. Further, within the study by Young et al. (2017) [34], it was found that older adults were less likely to adhere to wearing the technology. Finding methods to improve older adults’ perceptions of technology is important. One theoretical approach in the literature is the Technological Acceptance Model (TAM) [51]. The TAM outlines that the users’ perceived ease of use and perceived usefulness are key factors in whether they will adopt the technology. Based on the TAM, finding methods to alter cardiac patients’ health beliefs and to demonstrate the accuracy and the security of the technology are some methods to increase adoption [52]. 

### 4.3. Effects of the Technology

The third aim was to investigate the effects of connected health technologies on a range of health-related outcomes. The rationale for implementing home-based technologies is primarily to save healthcare costs, increase accessibility to services and to improve patient health [53,54]. Several positive effects were observed within the included studies. Hospital readmission and service utilisation are important outcomes in terms of cost-effectiveness and have large implications for those living in rural areas [30]. In several of the studies, the group using home-based technology reduced hospital readmission rates in comparison to comparator groups [35,37]. This is hypothesised to be achieved through improvements in the patient’s self-management, which can be mediated by the technology [5]. There were also several health outcomes which were improved within the technology groups. Lear et al. (2014) [36] found decreased cholesterol, saturated fat, and low-density lipoproteins in the virtual cardiac rehabilitation group. Moreover, the virtual cardiac rehabilitation group increased their maximal treadmill time test in comparison to the usual care group. With regard to how technology generates benefits to participant health, it can create positive behaviour change to physical activity through goal setting, prompts and cues, self-monitoring behaviour and social support [55]. It should also be considered that virtual forms of healthcare provide increased access to professional support without the need to travel long distances. However, there are some important considerations of implementing technology within the context of individuals living in rural areas. 

Despite all participants residing in primarily rural locations, not all rural communities are homogenous [30]. Differences in rural communities can range in infrastructure/accessibility, demographics, connections, proximity to healthcare facilities and climate/weather [7]. Within cardiac patients living in rural areas, home exercise with integrated technology is one method of improving health outcomes, however, based on the lack of studies which met the inclusion criteria, more efforts need to be made to increase research on home-based technology in rural areas [15]. Connected health technologies appear to be the future of healthcare, and promoting self-management [56] and identifying methods to increase patient adherence and acceptability requires attention. Although there are mixed findings, the systematic review overall shows positive effects of connected health technology on health-related outcomes and patient experiences. Whilst the use of connected health technology was warranted, there are barriers to implementation. Findings regarding adopting digital technology in healthcare show factors such as resistance to change, perceived ease of use and digital literacy and age to be barriers [57]. Moreover, in many rural locations, connectivity in terms of Wi-Fi and 4G/5G are substandard and pose further barriers to implementing telehealth [58]. As technology continues to advance, it is anticipated that the digital infrastructure in rural areas will begin to mirror that seen in urban areas [58]. 

### 4.4. Limitations

There were several limitations to the current systematic review that warrant discussion. In two of the included studies, there was no randomisation, which reduces the overall quality of the research. Initially, it was planned that a meta-analysis would be conducted, however, due to the heterogeneity of outcome measures and forms of technology, pooling of results was not possible. Within the studies, there were eight different types of technology implemented, and this made it challenging to confirm which form of technology resulted in the positive findings. Finally, there were a range of rural areas included within the review from different locations worldwide. Previous research has shown a disparity between rural areas with regard to physical activity, and thus what works well for one area may not be useful for another [7].

### 4.5. Future Research

As highlighted by Harrington et al. (2020) [9] and confirmed by the findings of the current review, future research should seek to implement new forms of telehealth and wearable technology to continue to address the barriers experienced by patients living in rural areas. The current review demonstrates a lack of studies that have implemented wearable technology, smartphone applications or videoconferencing software in rural cohorts, all which have been shown to have positive effects in other contexts. Such study designs should include randomisation, control groups and large sample sizes, to provide generalisable research where firm conclusions can be drawn on the effectiveness of specific types of technology. However, one drawback to conducting research on specific forms of wearable technology is the rate at which new devices are released, and thus technology evolves at such a rate that it is challenging for research to keep up. With only one study from the five measuring physiological parameters, future research should consider such outcomes. Moreover, another avenue of research regarding technology may be identifying methods to increase adherence and attitudes towards the technology. Previous research [56] has highlighted that the uses and benefits of many forms of technology are short lived and suggested that behaviour change techniques and incentives may address this. Long-term studies should aim to demonstrate how the effects of different forms of technology are altered over time.

## 5. Conclusions

Due to the low number of studies which met the inclusion criteria of the systematic review, it is clear there is a lack of research investigating the use of digital technology to enhance the health outcomes of individuals living with CVD in rural areas. Individuals living in rural areas have increased barriers to attending healthcare settings and conducting physical activity. Living with a form of CVD requires both regular physical activity and healthcare to effectively manage the condition. Based on the findings of this review, connected technology may be one method to reduce geographical inequalities. Specifically, wearable technology has been underutilised despite the potential benefits to increasing physical activity levels. Acceptability and adherence to the technologies in the review were mixed, highlighting the need for better methods to increase patients’ understanding in how to use the technologies and perhaps the development of more user-friendly technologies or more ‘wearable’ wearables. The primary aim of implementing technology is to save costs to healthcare services, increase accessibility and communication, resulting in better health outcomes and more uptake. The evidence within this review demonstrated some positive outcomes in relation to reduced healthcare utilisation and CVD risk factors in addition to improved self-management. It appears connected health technologies can be useful in rural populations with CVD, however as technology continues to advance, more research is required assessing both technology literacy of rural patients and developing ways to increase this.

## Figures and Tables

**Figure 1 ijerph-19-02302-f001:**
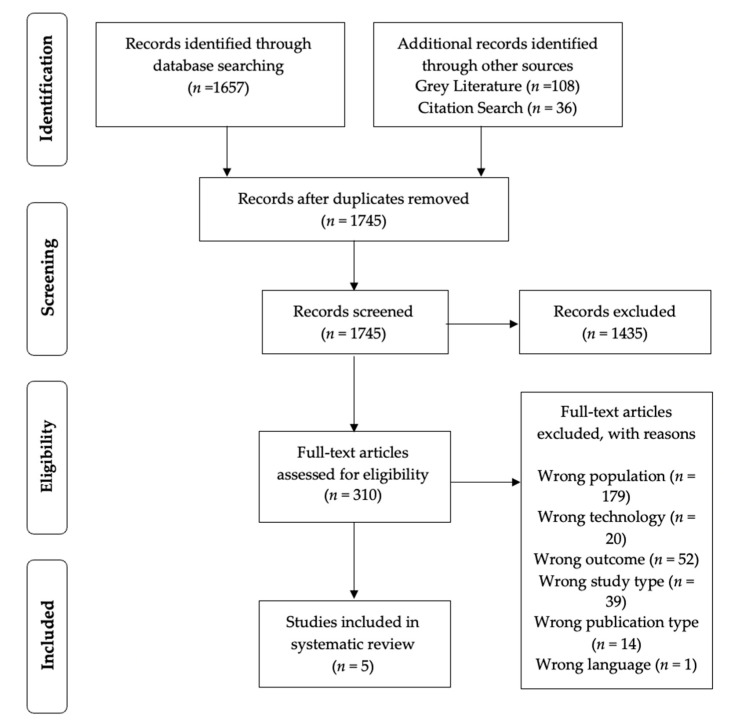
Flow Diagram of search and inclusion process for identification of articles.

**Table 1 ijerph-19-02302-t001:** Study characteristics.

Author and Country	Study Design	Cardiac Issue	Sample Size	Age (Years)	Type of Technology
**Lefler et al.** [33] (Southern USA)	Mixed methods	Heart failure	28 total 16 males 12 females	55–59: 5 60–64: 6 65–69: 7 >70: 10	Android tablet Bluetooth body weight scale Wrist worn BP monitor
**Young et al.** [34] (Nebraska, USA)	Secondary analysis	Heart failure	100 total 64 females 36 males	70.2 (±12.21)	Actigraph GT3X-BT accelerometer
**Riley et al.** [35] (Arizona, USA)	Programme evaluation	Heart failure	147 Enrolled group (22 female 23 male) Matched group (22 female, 23 male) Declined group (21 female, 36 male)	Enrolled group (66.0 ± 14.5) Matched group (65.9 ± 14.7) Declined group (66.3 ± 14.1)	Mobile health remote monitoring kit Blood pressure monitor Heart rate monitor Pulse oximetry Bluetooth weight scale
**Lear et al.** [36] (Vancouver, Canada)	Randomised, controlled trial (Pre/post)	Myocardial infraction, Coronary artery bypass graft, Angioplasty, other.	78 (71 completed the study) 66 males and 12 females	Usual care group: 58.4 (52.8–64.7) Intervention group: 61.7 (51.5–65.2)	Heart rate monitor (Polar S610i) Blood pressure monitor (Lifesource UA779) One to one chat Internet tasks
**Huntington et al.** [37] (South Dakota, USA)	Prospective, non-randomized, two-centre pilot project	Congestive heart failure	250 (98 enrolled, 152 non-enrolled) 55% male enrolled, 48% male non-enrolled	18–40 (7) 41–55 (14) 56–70 (63) 71–85 (123) >85 years (43)	Education Telephone

BP: Blood pressure.

**Table 2 ijerph-19-02302-t002:** Intervention characteristics.

Author and Year	Intervention Description	Outcome Measures	Duration	Key Findings
**Lefler et al., 2018** [33]	mHealth group took daily readings of BP and weight sent to nurses, with a weekly phone call on symptom status. ‘In-home group’ did the same without nurse assistance. ‘Standard of care’ group did not receive equipment and were told to follow regular care instructions.	Perspectives, self-care, communication and engagement, adherence, technology usability. Health status, emergency department visits.	12 weeks	Good communication and engagement (80–92% satisfaction). 100% of participants in intervention groups reported not forgetting to daily monitor symptoms. 12/15 participants ranked the equipment as usable. 93% of participants agreed the equipment was easy to use. No difference in ED visits were found between the groups. 20% of the participants in the home equipment group noted changes in BP or weight over 12 weeks.
**Young et al., 2017** [34]	Secondary analysis of a 12-week RCT aiming to improve HF self-management. Participants were required to wear an accelerometer for a minimum of 8 h for 7 consecutive days.	Wear time, acceptability, reliability. Working status, medications, utilisation of healthcare services (hospitalisations)	12 weeks	Wear time (15.7 h ± 3.3 h weekdays, 15.8 h ± 3.7 h weekends). 54% of participants had 100% compliance. Discomfort, fit issues (too tight/loose), skin issues (sweating, rashes), interreference with daily activities and difficulties removing/putting on were cited as barriers to using the device.
**Riley et al., 2015** [35]	Self-monitoring intervention. Participants were instructed to collect daily measurements for 3–6 months. Review the data before submitting to the care coordinator.	Rehospitalisation and health care utilisations, satisfaction, usability. Cost effectiveness of intervention.	6 months	Intervention group significantly reduced their healthcare utilisation across all time frames. Participants noted the technology was easy to use, the majority provided usability ratings of 4/5. Average hospital charges decreased from $129,480 to $36,914.
**Lear et al., 2014** [36]	VCRP intervention Attend a 16-week VCR programme. Complete risk factor and lifestyle forms, one to one chat sessions, weekly education online. Wear HR monitor during exercise twice a week, BP pre/post exercise.	Exercise capacity, cholesterol, blood glucose, BMI, leisure time physical activity, diet, hospital admissions, emergency room admissions	12 week pilot study 4 months of VCRP 12 month follow up	Intervention group increased their treadmill time by 45.7 s compared to the usual care over the 16months. Cholesterol was lower in the intervention group a follow up. 22 adverse events in usual care group compared to 8 in intervention group. VCRP was seen to be accessible, effective, and convenient via interviews.
**Huntington et al.,** **2013** [37]	Education and follow-up intervention. Participants in the enrolled group received four educational calls from nurses and final call 30-days post discharge.	30-day all cause hospital readmissions.	12 months	Significant 42% relative reduction in 30-day readmission rate participants in the pilot program. Economic savings from attending the pilot programme.

BP: blood pressure; HR: heart rate; VCRP: virtual cardiac rehabilitation programme.

**Table 3 ijerph-19-02302-t003:** Quality of the included studies.

Author	Reporting (10)	External Validity (3)	Internal Validity Bias (7)	Confounding (6)	Power (1)	Total (28)
[34]	10	2	4	5	0	21
[33]	6	2	5	4	0	17
[36]	10	2	7	5	1	25
[37]	8	1	5	3	0	17
[35]	10	2	4	4	0	20

## Data Availability

Systematic review extra data is Appendix A.

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
