# Peer review of "Does Connected Health Technology Improve Health-Related Outcomes in Rural Cardiac Populations? Systematic Review Narrative Synthesis"

_ijerph, 2022, doi:10.3390/ijerph19042302_

Round 1

Reviewer 1 Report

Individuals living in rural areas are more likely to experience cardiovascular diseases (CVD) and have increased barriers to regular physical activity in comparison to those in urban areas. This systematic review aimed to understand the types and effects of home-based connected health technologies, used by individuals living in rural areas with CVD.

There was a decisive drawback in this study, because meta-analysis could not be performed due to too few studies. Therefore, The usefulness of connected health technology could not be presented practically and intuitively. Nevertheless, the authors systematically presented the purpose and results of the study through commonly used methodologies. In addition, this study covers what is consistent with Special issue.

The purpose of this study was to examine the types and effects of connected health technology studied in rural areas. The authors selected people living in rural areas, a vulnerable group that has been rarely studied so far, and systematically described the necessity and limitations of the research targeting them.  

The authors tried to systematically review and describe the rural-based intervention studies, which are somewhat lacking.  

Recent review articles perform meta-analysis. The authors should have collected as many connected health technology studies as possible in the initial study design and presented statistical values for their impact on health-related outcomes. Then, the authors should have done a subgroup analysis of people in rural areas. However, this study lacked that analysis.

The authors sought to systematically describe the types and effects of connected health technology by citing limited studies. Overall, the study's conclusions are in line with the evidence and arguments presented by the authors. 

The references are appropriate.

It is recommended to present one figure using graphic art that intuitively shows the technique type, target sample, duration, and key findings introduced in the five studies.

Author Response

February 13, 2022

RE: ijerph- 1596313

Anonymous Reviewer #1,

Thank you very much for reviewing our manuscript. Your comments and suggestions are greatly appreciated. We have carried out the suggested changes and revised the manuscript accordingly. The suggestion regarding the collection of all connected health technologies and their effect on health-related outcomes is a good suggestion and one we may look to investigate in a meta-analysis in the future.

I appreciate all the comments from the reviewer, these have helped improve the quality of the manuscript.

Thank you for your response and your time.

Yours Sincerely,

Matthew Fraser.

Reviewer: It is recommended to present one figure using graphic art that intuitively shows the technique type, target sample, duration, and key findings introduced in the five studies.

Response: Thank you for your suggestion, we have created a graphical abstract for approval by the editor which includes the target sample, intervention characteristics, results, and findings.

Reviewer 2 Report

This systematic review seeks to examine the impact of health technology (e.g. mHealth and wearables) on health outcomes for patients with cardiovascular disease. I appreciate that the authors were upfront that their originally intentions were to conduct a systematic review of cardiac rehabilitation in rural populations, but expanded the disease criteria after initial searches yielded zero results, and also that they intended to do a meta-analyses, but could not due to the heterogeneity of the studies. After screening the articles, only five articles met all study inclusion criteria. The authors provide a narrative synthesis of their findings. Overall, I thougth they did an excellent job on this narrative synthesis as it provided insight into what has been done in this area, what the outcomes were, and other considerations including adherence and hesitancy toward the technology. The methods were clear as well, and the authors provided their search terms in the appendix. There search strategy was comprehensive as they examined 9 of the most common search databases including PubMed, Web of Science, Cinahl, and Google Scholar, as well as several government and key organization sites for grey literature. The introduction provided a strong rationale for the study, and the discussion adequately wrapped up the results and commented on the challenges of studying this technology because today's products become obsolete in just a few years. I only have a few minor comments.

1) I found the abbreviation of physical activity as PA distracting. I think it would be better to just write the full term out throughout. 

2) Can the authors clarify a few things in the exclusion criteria:
    a) were most of the 126 articles removed for "wrong population" because they were not in rural populations?
    b) what were the 18 studies removed for wrong study design. In the PICOs criteria on lines 181-183 it says both randomized and non-randomized studies were included, but no other criteria was mentioned that would warrant exclusion.

Author Response

February 13, 2022

RE: ijerph- 1596313

Anonymous Reviewer #2,

Thank you very much for reviewing our manuscript. Your comments and suggestions are greatly appreciated. We have carried out the changes that were suggested and revised the manuscript accordingly.

I appreciate all the comments from the reviewer, these have helped improve the quality of the manuscript.

Thank you for your response and your time.

Yours Sincerely,

Matthew Fraser.

Reviewer: I found the abbreviation of physical activity as PA distracting. I think it would be better to just write the full term out throughout.

Response: All ‘PA’ have been changed to ‘physical activity’.

Reviewer: were most of the 126 articles removed for "wrong population" because they were not in rural populations?

Response: Not being from a rural population was the primary reasons why these 126 articles didn’t meet eligibility.

Reviewer: What were the 18 studies removed for wrong study design. In the PICOs criteria on lines 181-183 it says both randomized and non-randomized studies were included, but no other criteria was mentioned that would warrant exclusion.

Response: The term ‘wrong study design’ was used to categorise a range of types of articles. Within the 18 studies removed, some of the reasons were ‘published study protocol’, ‘review article’, ‘did not conduct a study (discussion article, letter to the editor, perspectives, scientific statement)’, ‘cross-sectional study with no comparators used’.